# A Comprehensive Analysis of *Echinococcus granulosus* Infections in Children and Adolescents: Results of a 7-Year Retrospective Study and Literature Review

**DOI:** 10.3390/pathogens14010053

**Published:** 2025-01-10

**Authors:** Cristina Maria Mihai, Ancuta Lupu, Tatiana Chisnoiu, Adriana Luminita Balasa, Ginel Baciu, Vasile Valeriu Lupu, Violeta Popovici, Felicia Suciu, Florin-Daniel Enache, Simona Claudia Cambrea, Ramona Mihaela Stoicescu

**Affiliations:** 1Department of Pediatrics, Faculty of General Medicine, “Ovidius” University of Constanta, 900470 Constanta, Romania; cristina_mihai@365.univ-ovidius.ro (C.M.M.); adriana.balasa@365.univ-ovidius.ro (A.L.B.); 2Pediatrics, County Clinical Emergency Hospital of Constanta, 900591 Constanta, Romania; 3Department of Pediatrics, “Grigore T. Popa” University of Medicine and Pharmacy, 700115 Iasi, Romania; ancuta.ignat1@umfiasi.ro (A.L.); vasile.lupu@umfiasi.ro (V.V.L.); 4Department of Pediatrics, Faculty of Medicine and Pharmacy, “Dunărea de Jos” University of Galati, 800008 Galati, Romania; ginelbaciu@yahoo.com; 5Center for Mountain Economics, “Costin C. Kritescu” National Institute of Economic Research (INCE-CEMONT), Romanian Academy, 725700 Vatra-Dornei, Romania; 6Department of Analysis and Quality Control of Drugs, Faculty of Pharmacy, “Ovidius” University of Constanta, Str. Căpitan Aviator Al. Șerbănescu, nr.6, Campus Corp C, 900470 Constanta, Romania; felicia.suciu@365.univ-ovidius.ro; 7Department of Pediatric Surgery and Orthopedics, Faculty of General Medicine, “Ovidius” University of Constanta, 900470 Constanta, Romania; dr.enache@chirurgiecopii.ro; 8Pediatric Surgery, County Clinical Emergency Hospital of Constanta, 900591 Constanta, Romania; 9Department of Infectious Diseases, Faculty of General Medicine, “Ovidius” University of Constanta, 900470 Constanta, Romania; cambrea.claudia@gmail.com; 10Department of Microbiology and Immunology, Faculty of Pharmacy, “Ovidius” University of Constanta, Str. Căpitan Aviator Al. Șerbănescu, nr.6, Campus Corp C, 900470 Constanta, Romania; ramona.stoicescu@univ-ovidius.ro

**Keywords:** *Echinococcus granulosus sensu lato*, hydatid cyst, Romanian pediatric patients, liver and lung involvement, medical and surgical treatment, IgG, IgE, eosinophilia

## Abstract

Cystic echinococcosis (CE) is a neglected tropical parasitic disease linked with significant social and economic burdens worldwide. The scientific community has minimal information on echinococcosis in Romanian people, and hospital medical records are the only sources that may be used to investigate its status. A 7-year retrospective clinical study on pediatric patients with CE from Southeast Romania was performed, and 39 children and adolescents were included, aged 2–15 years old. They were hospitalized with cystic echinococcosis in the Pediatric Department and Pediatric Surgery Department of Constanta County Clinical Emergency Hospital “St. Apostle Andrew” between 1 January 2017 and 1 October 2024. Twenty-nine (74.36%) pediatric patients came from rural zones, and 10 (25.64%) had urban residences. In total, 28 children (71.79%) had contact with four different animals (dogs, goats, pigs, and sheep); only four were from urban zones, and they had contact only with dogs. Data regarding the length of hospital stay, cyst location, and complications were collected and analyzed. According to the medical files, the diagnosis was established using imaging techniques and serological tests for CE. IgE and IgG reported appreciable variations in correlation with all parameters, and significant differences (*p* < 0.05) were recorded. IgE levels considerably increased in cases of no animal contact, pulmonary involvement, complications, surgical treatment, and multiple hospitalizations. Moderate IgE values were recorded in cases of urban residences, pig and sheep contact, and hepatic involvement. The IgG concentration considerably increased with sheep contact and moderately increased in cases of rural zones, hepatic involvement, complications, and surgical treatment. The results show that incidental discovery, symptoms, complications, multiple dissemination, pulmonary involvement, and dog and pig contact increase the hospitalization time. Extensive data analysis supports our results. Our findings highlight the complexity of managing *E. granulosus* infections in children and evidence the importance of a multidisciplinary approach, combining early diagnostic tools, tailored medical therapy, and careful surgical intervention when necessary.

## 1. Introduction

The parasitic zoonotic infection known as human cystic echinococcosis (CE) is brought on by the larval stages of *Echinococcus granulosus sensu lato* [1]. *E. granulosus*, the causative agent of CE, is a tapeworm parasite transmitted from animals to humans, commonly affecting children in endemic areas. This parasitic infection leads to the formation of cysts, primarily in the liver and lungs, which can cause severe complications if it is untreated [2]. Cystic echinococcosis has been recognized since 1950 as a public health problem [3]. Echinococcosis belongs to the list of 17 neglected tropical diseases (NTDs); WHO advocates concerted control efforts because CE is a prioritized neglected zoonotic disease [4].

*E. granulosus* can be considered a cosmopolitan parasite, being found on all populated continents. The disease is widespread, especially in countries where agriculture, mainly sheep farming, occupies a bare place in the national economy, with a higher incidence recorded in the areas where hygiene and development standards are relatively low [5]. Globalization, including migration and tourism in recent years, has increased the interest in this pathology in countries with a low *Echinococcus granulosus sensu lato* infection or where the disease was considered eradicated [6,7,8,9].

Numerous elements, such as education, agriculture, culture, socioeconomic status, and the environment, are essential in the disease’s spread [10]. According to research on domestic animals from rural areas (sheep, goats, pigs, and dogs), up to 34% were found sick [11,12,13]; large quantities of *Echinococcus* eggs are transferred from animals to water, vegetables, or farmlands, being contagious for months [14]. Canids are the final host in the life cycle of *E. granulosus*, while ungulates are mainly the intermediate host [2,12]. As unintentional hosts, humans contract the illness via contact with dogs, contaminated food or water consumption, or hand-to-mouth transmission of the parasite’s eggs [14]. The oncospheres are released from the embryonated eggs through the action of stomach juices after ingestion. The hexacanth embryos cling to and penetrate the enteric mucosa in the small intestine. The blood flows through capillaries and then carries them to other organs [15,16]. One or more hydatid cysts grow in the organs due to the infection [17,18]. Various patterns of hydatid cysts and multiple organ involvement have often been observed in children because of the undeveloped filtering mechanisms of the liver and lungs throughout infancy and childhood, as well as the role of intestinal lymphatic vessels in parasite transmission [19]. Usually, cysts develop in the liver (70%) and lungs (20%), yet they can also be found in other organs like the brain, bones, heart, and urinary and genital tract [20,21]. The liver and lungs are primary targets for hydatid cyst formation due to their filtration role in systemic circulation [22]. Children rarely manifest echinococcosis due to age. This disease, which primarily affects the liver, is often not included in differential diagnoses of abdominal diseases, and missing the diagnosis can have devastating consequences. Symptomatology in its early stages is mild or absent [23]. When complications occur, the symptomatology depends on the cyst’s stage, number, localization, and size [23]. Nonspecific symptoms such as fever, chest discomfort, coughing, and dyspnea can indicate pulmonary involvement [24]. A pulmonary cyst is frequently found by accident on a chest X-ray or may be found due to respiratory symptoms after cyst rupture or cyst infection. Simple cysts, inflammatory masses, and benign or malignant tumors are among the many differential diagnoses for pulmonary cysts in children [22,23,24,25]. Multiple investigations are necessary for a reliable diagnosis, including histology, serology, imaging, and the polymerase chain reaction (PCR) approach [26,27,28].

The liver is the most common site for *E. multilocularis* infection, with over 90% of cases affecting it, typically in the right lobe [29]. Cystic lesions grow slowly, causing pressure or discomfort in older children and teens, particularly in the right hypochondrium [19]. Jaundice is rare, but cholestasis, jaundice, or portal vein thrombosis/portal hypertension can occur, especially if the infection affects the liver hilum [30]. Diagnosis of CE relies on magnification techniques, and immunoblotting (IB) and enzyme-linked immunosorbent assays (ELISAs) are key diagnostic tools. *E. granulosus* infection in children presents unique challenges, and timely diagnosis and treatment are crucial in endemic areas [8,9,21,31].

With 200,000 new cases per year previously identified worldwide, human CE accounts for over 95% of the estimated 2–3 million cases [32,33]. However, because of the high expense of treatment, missed income, and productivity diminution linked to cattle, CE is a significant cause of healthcare and economic losses in many world regions. CE is endemic in EU countries like Spain, Italy, Bulgaria, Greece, and Romania and in non-EU countries such as Macedonia, Moldova, Serbia, and its neighboring countries [32,33].

In Romania, 52.16% of the population lives in urban environments, while the rural population represents 47.83% [34]. The incidence of CE was 5.6 per 100,000 inhabitants/year, according to the studies conducted by Lupașcu et al. between 1953 and 1963 [35]. Between 1991 and 1995, 1000 new cases per year were recorded, according to Ciuca et al., with an increased incidence in sheep farming zones (Sibiu, Dobrogea areas) [36].

A 25-year retrospective study in Western Romania analyzed 144 pediatric patients with cystic echinococcosis. The findings revealed that 58.3% of the patients were from rural areas, with the number of cases increasing with age—from 9% in the 3–5 age group to 59.7% in the 11–17 age group. The liver was the most frequently affected organ (65.3%), and a significant association between gender and the affected organ was noted; liver cysts were more commonly diagnosed in girls, while lung cysts were recorded primarily in boys [37].

In 2022, the estimated population in Constanta was 655,997 inhabitants [34]. Rural residents accounted for 34.01% of the population in Constanta County. The regional continental climate with Mediterranean influences–warm summers and mild winters–is ideal for developing technological plants and grains, and zonal geography with a predominant plain is favorable for sheep raising. The scientific community has minimal information on echinococcosis in Romanian people [37,38,39] because Romania has no national regulations to manage *E. granulosus*, and reporting new cases is not mandatory [40,41]. Hospital medical records are the only sources that may be used to investigate CE status in our country. Therefore, the present study aims to enrich the scientific database with concrete information regarding pediatric echinococcosis in Southeast Romania, based on data collected from Constanta County Clinical Emergency Hospital “St. Apostle Andrew”.

## 2. Materials and Methods

According to the Ethical Committee of the Ovidius University of Constanta, Faculty of Medicine (protocol 21 approved on 15 November 2024), a retrospective clinical study was performed on pediatric patients with CE from Southeast Romania.

The 39 children and adolescents included in the study were 2–15 years old. They were hospitalized with cystic echinococcosis in the Pediatric and Pediatric Surgery Departments of Constanta County Clinical Emergency Hospital “St. Apostle Andrew” between 1 January 2017 and 1 October 2024.

Hospital databases and medical charts were reviewed. Data regarding age, gender, area of residence, length of hospital stay, cyst location, and complications were collected and analyzed. According to the medical files, the diagnosis was established using imaging techniques (radiography, ultrasonography, computed tomography) and serological tests for CE (*E. granulosus* antibodies IgG, using the ELISA method). A negative result of antibodies does not exclude a hydatid; the test may indicate negative or equivocal results due to the low antibody level in the early stages of the infection. If there is a clinical suspicion, the test must be repeated after 2–4 weeks. Significant cross-reactivity with *Taenia solium* has been reported. A positive result does not exclude the influence of other pathogens [42,43].

### Data Analysis

The extensive data analysis used different tools of XLSTAT Life Sciences v 2024.3.0. 1423 by Lumivero (Denver, CO, USA): descriptive analysis, ANOVA single factor, Kruskal–Wallis analysis, correlations between variable parameters, and heat maps [44]. Following the descriptive statistics, the variable parameters are displayed as absolute frequencies (number, N) and relative frequencies (percentage) [45]. Statistical significance was established at *p* < 0.05 [46].

## 3. Results

### 3.1. Sociodemographic and Epidemiological Characterization of Pediatric Patients

Data are registered in Table 1. Statistically significant differences are considered between both groups with rural and urban residences (*p* < 0.05).

*Echinococcus* sp. transmission was investigated, including residence in rural and urban zones, sex, age, potential contact with animals, and animal type. Therefore, Table 1 shows that 29 (74.36%) pediatric patients came from rural zones, and 10 (25.64%) had urban residences. Most were boys (24/39, 61.54%), while the 10–15 age group recorded the highest incidence (24/39, 61.54%). Twenty-eight children (71.79%) had contact with animals (dogs and ungulates: goats, pigs, and sheep). Animal contact was predominant in rural zones in 24/29 (82.76) patients. The children from rural regions had contact with various animals: goats (9/24, 37.50%), sheep (9/24, 37.50%), pigs (5/24, 20.83%), and dogs (1/24, 4.17%). Only 40% of pediatric patients (4/10) with urban residences had contact exclusively with dogs.

### 3.2. Main Symptoms at Presentation and Organ Involvement

These findings are presented in Table 2, grouped by hospitalization period (days).

The illness was incidentally discovered in six children (15.38%): 5/29 (17.24%) from the rural zones and 1/10 (10%) from the urban ones.

Thirty-three children (84.62%) manifested various general and specific symptoms at presentation in the Emergency Care Unit (ECU). Sixteen children (41.03%) had abdominal pain (AP) alone (9/39, 23.08%) or associated with one (AP +1, 2.56%) or two symptoms (AP +2, 15.38%): fever, headache, nausea, or vomiting. Eleven children (28.21%) had a cough (combined with one (Cough +1, 17.95%) or two (Cough +2, 10.26%) different symptoms: fever, hemoptysis, thoracic pain, thoracic back pain, or shortness of breath. Six children (15.38%) had other unusual symptoms (Other 1), such as axillary adenopathy, back pain, left thoracic pain, tachycardia, or vomiting alone (Table 2, Figure 1 and Figure 2A)

The main symptoms at the presentation in the Emergency Care Unit could help identify the organ involved in CE (Figure 1) and eventually multiple organ dissemination (Figure 2A).

Figure 1 shows that AP (37.50%, 9/39), AP +1 (4.17%, 1/39), and AP +2 (20.83%, 5/39) were essential symptoms of liver involvement. Cough +2 and unusual symptoms (Other 1) were detected in 16.67% (4/39) of pediatric patients, while Cough +1 was absent (Figure 1). In pulmonary hydatidosis, the patients mainly had Cough +1 (40.00%, 6/39) and Cough +2 (26.67%, 4/39); AP +2 and Other 1 were detected in four (6.67%) and two (13.33%) children, respectively (Figure 1). Four children with hepatic hydatidosis (16.67%) and two with pulmonary hydatidosis (13.33%) were incidentally discovered (Figure 1).

Seven children were detected with multiple organ diffusion (MD, Table 2). Figure 2A shows that only four categories of severe symptoms were involved in MD: AP +2 (14.29%, 1/39), Cough +1 and +2, and Other 1 in equal measure (28.57%, 2/29).

Liver involvement was detected in 24 children (61.54%): 15 (38.46%) had lungs affected, and seven children (17.95%) were detected with multiple organ diffusion of CE (Table 2 and Figure 2B). Figure 2B reveals that MD is mainly present in pulmonary CE (71.43%, 5/39) as opposed to hepatic CE (28.57%, 2/39).

Eighteen children (46.15%) had various complications (Table 2 and Figure 2C). The complication rate is significant when MD exists: 71.43% of children with MD (5/7) have complications (Figure 2C).

The main complications are biliary fistula (16.67%), hepatic hydatid cyst alone or associated with pneumonia, and pleural effusion (11.11%), as illustrated in Figure 2D. Other complications have lower incidence: bronchopneumonia, chronic pancreatitis, biliary fistula associated with hepatic hydatid cyst infection or rupture, peritoneal abscess, pulmonary hydatid cyst with/without infection, pleural fistula with pneumothorax (5.56%, Figure 2D). Hepatic and pulmonary CE are associated with biliary fistula (20% vs. 12.50%), hepatic hydatid cyst with/without pneumonia, and pleural effusion (10% vs 12.50%). Hepatic involvement is exclusively associated with the following complications: chronic pancreatitis (10%), biliary fistula with hepatic hydatid cyst infection (10%) and rupture (10%), peritoneal abscess (10%), and pulmonary hydatid cyst (10%). Pulmonary involvement is only associated with bronchopneumonia (12.50%), hepatic hydatid cyst (12.50%), pleural fistula with pneumothorax (12.50%), and pulmonary hydatid cyst infection (12.50%, Figure 2D).

The hospitalization period varied from 3 to 30 days. Twenty-two children (56.41%) were hospitalized for 6–10 days, 9/39 (23.08%) for 3–5 days, 5/39 (12.82%) for 11–15 days, and 3/39 (7.69%) for 20–30 days (Table 2).

A Pearson correlation was applied to all data recorded belonging to this sequence and showed that AP +1, AP +2, Cough +1, and Other 1 symptoms, as well as both organ involvement, are highly correlated with complications (r = 0.965–0.999, *p* < 0.05, Figure 3).

Data from Figure 3 also reveal that 6–10 days of hospitalization strongly correlates with liver involvement, multiorgan diffusion, severe symptoms (AP +1, AP +2, Cough +1, Other 1), and the presence of complications (r = 0.965–0.999, *p* < 0.05).

### 3.3. Laboratory Analyses, Evolution, and Treatment

The data recorded in Table 2 show that 32 pediatric patients (82.05%) had IgE > 100 i.u./mL, 31/39 (79.49%) had IgG > 1.1 i.u./mL, and 13/39 (33.33%) had eosinophilia >0.5/µL. Eleven CE patients (28.21%) received medication (albendazole); the dosage used in children was adjusted according to body weight, with a standard dosage of 10–15 mg/kg/day divided into two daily doses. Surgical intervention was necessary for 28 patients (71.79%). In 18 cases (46.15%), multiple hospitalizations were required.

The principal component analysis illustrated in Figure 4 is based on the Pearson correlation, which evidences a substantial correlation between the high values of all three parameters (r = 0.964–0.997, *p* < 0.05).

They also significantly correlate with liver involvement and complications (r = 0.973–0.997, *p* < 0.05), TS (r = 0.968–0.999, *p* < 0.05), 6–10 days of hospitalization, and MH-yes (r = 0.954–0.998, *p* < 0.05). IgE > 100 i.u./mL and IgG > 1.1 i.u./mL also highly correlate with lung involvement and multiple organ diffusion (r = 0.961–0.997, *p* < 0.05). Multiple hospitalizations substantially correlate with 6–10 days of hospitalization, TS, complications, increased Eos, IgG, and IgE values, liver involvement, and multiple organ diffusion (r = 0.961- 0.997, *p* < 0.05).

Therefore, all three parameters are considerably associated with all aspects of CE; they are essential biomarkers for CE evolution, treatment, and prognosis.

## 4. Discussion

CE is globally widespread, causing significant public health and economic burdens, particularly in endemic regions, yet remains a neglected zoonotic disease [47,48,49,50,51,52,53,54,55,56,57,58,59]. In Romania, hydatidosis is the most significant helminthic zoonosis due to its severe clinical implications [60]. Our study corroborates previous findings, with 28 of 39 CE patients reporting contact with different animals, such as dogs, sheep, goats, or pigs. The liver is most frequently affected, followed by the lung; other tissues, including bone, can be affected [61,62,63,64,65,66,67,68,69,70,71,72,73]. The higher incidence of liver involvement in pediatric patients may be attributed to the larger relative size of the liver in children and its role as a primary filter for bloodstream pathogens [74,75].

### 4.1. Correlation Between Sociodemographic Data and Clinical Findings

The whole findings of the present study are summarized in Table 3.

#### 4.1.1. Children vs. Adolescents

Hepatic involvement was recorded in 16 adolescents aged 10–15 years (66.67%), 7 patients aged 4–9 years (63.64%), and 1 aged ≤3 years (25%, *p* < 0.05). Multiple organ diffusion was detected in the first two age groups: 20.83% (5/24) of adolescents and 18.28% (2/11) aged 4–9 years. In total, 25% of children ≤3 years were incidentally discovered (1/4). All pediatric patients had high levels of IgG and IgE (over 75%) and low eosinophil concentration (over 60%). The highest hospitalization period (20–30 days) was recorded in the lowest age group (≤3 years), of 25%, compared to 9.09% in those aged 4–9 years and 4.17% in adolescents (*p* < 0.05). Most complications occurred in adolescents (58.33%, *p* < 0.05), followed by children aged 4–9 years (27.27%) and ≤3 years (25%). In total, 79.17% of adolescents received surgical treatment (19/24), followed by 75% of children ≤3 years (3/4) and 54.55% (*p* < 0.05) aged 4–9 years (6/11). Multiple hospitalizations were requested for 75% of children ≤3 years (3/4), 54.17% of adolescents (13/24), and 18.18% aged 4–9 years (2/11), *p* < 0.05.

#### 4.1.2. Boys vs. Girls

Our findings showed that boys were more numerous than girls; from 39 patients, 61.53% were boys (24/39). Boys and girls presented similarities in hepatic involvement (62.50% vs. 60%), complication incidence (45.83% vs. 46.66%), and high levels of IgE and IgG (83.33% and 79.16% vs. 80%). Significant differences (*p* < 0.05) between boys and girls were observed in all three age groups (10–15 years (70.83% vs. 46.66%), 4–9 years (25.90% vs. 33.33%), and ≤3 years (4.16% vs. 20%)), residences (rural (66.66% vs. 86.66%) and urban (33.33% vs. 13.33%)), dog contact (16.66% vs. 6.66%), multiple organ diffusion (87.50% vs. 73.33%), surgical treatment (66.66% vs. 80%), 20–30 days of hospitalization (4.16% vs. 13.33%), and multiple hospitalizations (41.66% vs. 53.33%).

#### 4.1.3. Rural vs. Urban Residences

Our findings recorded substantial differences in pediatric patients from rural vs. urban zones. The incidence of contact with an animal was twice as high in rural zones (82.75% vs. 40%, *p* < 0.05), and ungulates were exclusively here (79.31% vs. 0%, *p* < 0.05); in urban zones, only contact with dogs was possible (3.44% vs. 40%, *p* < 0.05). Incidental discovery was higher in pediatric patients from rural zones (17.24% vs. 10%, *p* < 0.05). Considerable differences were recorded in organ involvement (liver (58.62% vs. 70%, *p* < 0.05) and lungs (41% vs. 30%, *p* < 0.05)) and multiple organ diffusion (20.69% vs. 10%, *p* < 0.05). There were also remarkable differences in the incidence of complications (48.27% vs. 40%, *p* < 0.05) and multiple hospitalizations (51.72% vs. 30%, *p* < 0.05). The incidence of IgE > 100 i.u./mL, surgical interventions, and high hospitalization period (20–30 days) was lower in pediatric patients from rural zones (79.31%, 68.96%, and 6.89% vs. 90%, 80%, and 10%).

However, most pediatric patients from both zones had high IgG levels (79.31% vs. 80%, *p* > 0.05).

#### 4.1.4. Contact with Different Animals

Animal contact (dog vs. ungulates vs. no contact) was significantly different in various age groups–10–15 years (60% vs. 52.17% vs. 81.82%, *p* < 0.05), 4–9 years (40% vs. 34.78% vs. 9.09%), and ≤3 years (0% vs. 13.04% vs. 9.09%, *p* < 0.05). It had a substantial impact on the main symptoms at presentation in ECU—Cough +2 and Other 1 (20% and 0% vs. 13.00% and 21.00% vs. 0%, 21.74%, and 9.09%, *p* < 0.05), multiple organ diffusion (0% vs. 26.09% vs. 9.09%, *p* < 0.05), and hospitalization period–3–5 days (40% vs. 17.39% vs. 27.27%, *p* < 0.05), 6–10 days (20% vs. 65.22% vs. 54.55%, *p* < 0.05), and 20–30 days (20% vs. 8.70% vs. 0%).

Significant differences between dog contact and ungulates were observed in AP +2 (20.00% vs. 13.04%, *p* < 0.05), incidental discovery (20.00% vs. 13.04%, *p* < 0.05), complication incidence (60.00% vs. 43.48%, *p* < 0.05), IgE > 100 (100% vs. 78,26%, *p* < 0.05), Eos > 0.5 (20.00% vs. 34.78%, *p* < 0.05), and treatment type–surgical intervention (80.00% vs. 65.22%, *p* < 0.05) and medication (20.00% vs. 34.78%, *p* < 0.05).

### 4.2. Essential Considerations

Our findings are consistent with global trends but underscore the need for heightened vigilance in endemic areas where multiorgan involvement can occur [76,77]. Most patients present a single lesion in a single organ [78], and our findings confirm it. Cystic echinococcosis becomes symptomatic in complications, such as a local compressive effect, fistulization in adjacent structures (biliary tree, bronchi), rupture with the spread of infection, or anaphylactic shock [79]. Frequently, patients are diagnosed during imaging investigations performed for other morbidities [80]. Eighteen children had various complications, while three of them were incidentally discovered. Our study also found that most pediatric patients with *E. granulosus* infections presented with nonspecific symptoms, such as abdominal pain, coughing, and fever. This aspect aligns with the existing literature, highlighting the difficulty in early diagnosis due to the often asymptomatic nature of hydatid cysts in the early stages [81,82,83].

There are several therapeutic alternatives accessible at the moment [84,85]. Smaller, simpler cysts (less than 5 cm) respond well to albendazole treatment [85,86]. However, medication by itself only removes 30% of cysts [87]. Eleven patients were treated with albendazole in the present study. In our study, data on the duration of albendazole administration before surgery were limited. However, for the cases where information was available, albendazole was administered preoperatively for a period ranging from 14 to 30 days. There were no recorded instances of cyst rupture during this period. Still, we acknowledge the need for further investigation into this aspect, given the ongoing controversy surrounding prolonged preoperative use and its potential risks. Albendazole administration aligns with current guidelines for pediatric patients and was carefully monitored to minimize adverse effects. For liver cysts >10 cm, those in danger of rupturing, and/or those that are difficult, surgery is recommended [88]. Laparoscopic surgery is seldom used, reducing postoperative abdominal infection risk. In biliary fistulas and commorbidities, the radical procedure (complete cystopericystectomy) is the preferred method [88,89]. Conservative methods are suitable when nonspecialist surgeons perform surgery in endemic areas. A novel approach is PAIR (puncture-aspiration-injection-reaspiration) as an alternative to surgery [90,91].

Twenty-eight patients had surgical treatment in the present study, and 12 had various complications. Combining medical and surgical approaches yields the best outcomes. Using albendazole as a preoperative and postoperative treatment reduced cyst viability and recurrence rates, highlighting its role as an essential adjunct to surgery. Notably, a subset of patients responded well to a conservative drug-only regimen, particularly those with small and uncomplicated cysts. Given the promising outcomes of albendazole treatment, further research into optimizing nonsurgical management strategies is warranted [92]. Invasive surgery might be avoidable in specific cases, thus reducing the morbidity associated with surgical interventions [93]. Moreover, it could be particularly beneficial in resource-limited settings where access to surgical facilities is restricted [94].

Eosinophilia describes an increase in eosinophils in the peripheral blood above the normal limit (greater than 500/mm^3^) [95]. It is frequently associated with parasitic diseases, especially those involving tissue parasites or larval invasion, but may also occur in other pathological contexts [96]. In parasitic diseases, eosinophilia is an essential marker of the innate and adaptive immune response [97]. Eosinophilia was reported in 20–34% of instances with hydatid cysts [98]. In cysts ruptures, leukocyte and eosinophil levels were greater [98]. In our study, eosinophilia was present in 33.33% of the cases, similar to other studies [99].

Parasites of the genus *Echinococcus* (most commonly *Echinococcus granulosus* and *E. multilocularis*) cause a complex immune response in which eosinophilia and IgE play a central role. Both components reflect the activation of the TH2-type immune response, which is characteristic of parasitic infections [100]. Eosinophilia and elevated IgE levels are essential markers in echinococcosis, reflecting activation of the host immune response against the parasite [39]. Although the correlation between the two mechanisms is well documented, the severity of these responses depends on the stage of the disease and the complex interaction between host and parasite [100]. Numerous studies have demonstrated that the location of the cyst, particularly in a vital organ like the liver, influences the immune response more than other affected organs in the body [101].

In our study, IgG levels refer specifically to antibodies targeting *Echinococcus granulosus*, which are used as markers for the presence of infection. These IgG antibodies are particular to the parasite and are considered a reliable diagnostic indicator. On the other hand, IgE levels are nonspecific and represent general allergic responses, which can be elevated in many parasitic infections, including CE. The unit used for IgG measurements is iu/mL; IgE is typically reported in iu/mL or ku/L. Regarding the elevated IgE antibody level in individuals with hydatid cysts in the liver as opposed to different organs, the influence of the hydatid cyst size was reflected in the IgG and IgE antibody levels [102]. In our study, IgE and IgG are highly related to complications, while Eos is poorly associated.

An essential aspect of the present research was evaluating immune responses post-treatment. High levels of specific IgG and IgE antibodies persisted in patients after cyst removal, suggesting a prolonged immune response that could be a marker for monitoring recurrence [103]. The immune system’s role in hydatid disease in children appears to be distinct from that in adults, necessitating further research into age-related immunological differences and their implications for treatment [27]. It could be particularly beneficial in resource-limited settings where access to surgical facilities is restricted. Understanding the immunological mechanisms may pave the way for new preventive measures, including vaccines [104,105,106].

Our study highlights the essential clinical and management aspects of cystic echinococcosis (CE) in children while underscoring the key differences compared to adults. These distinctions are critical for optimizing diagnosis and treatment strategies across age groups.

Children with CE frequently exhibit rapid cyst development, most likely due to their maturing immune systems and active metabolisms [27]. This fast development might cause symptoms early, usually as discomfort or pain in the abdomen, especially when the liver is involved [107]. Conversely, adults have a generally slower cyst formation and a more extended latency period before symptoms appear [108]. Extrahepatic involvement is more common in adult patients than in pediatric ones, and symptoms like obstructive jaundice or portal hypertension are commonly linked to the mass impact of the cysts. Additionally, the course of the disease varies significantly [109,110]. Cysts in children are more likely to rupture and have thinner walls, which may be caused by increased intra-cystic pressure [111]. This raises the possibility of serious side effects, including anaphylaxis or secondary infections. Indicative chronicity, comorbidities, and longer disease courses are more likely to cause portal vein thrombosis, biliary obstruction, and secondary bacterial infections [112].

Our study highlights the predominance of active cysts (CE1 and CE2) in pediatric patients, as classified by the WHO-IWGE system [113]. These findings are consistent with the early-stage presentation typically observed in children with shorter exposure to *Echinococcus granulosus* [114]. A smaller subset of cases involved transitional cysts (CE3), while no inactive cysts (CE4 and CE5) were identified in our cohort. This aligns with the existing evidence suggesting that inactive cysts are rarely encountered in children, likely due to the more acute nature of the infection in this age group.

Ultrasound was invaluable for diagnosing hepatic involvement in CE, offering high sensitivity and specificity for detecting and classifying cysts. Its noninvasive nature and real-time imaging make it ideal for pediatric patients, aiding diagnosis, disease progression monitoring, and treatment efficacy [115].

Our findings highlight that CE is a complex zoonotic illness that requires several approaches for diagnosis, management, and prevention. Professionals from various fields, such as physicians, surgeons, radiologists, infectious disease specialists, veterinarians, and public health specialists, must work together for effective management. Each offers a distinct area of expertise:Clinicians ensure a timely diagnosis and oversee medical management with antiparasitic drugs.Surgeons are essential for cases requiring cyst removal or the management of complications.Radiologists play a central role in diagnosing and monitoring cysts through imaging.Veterinarians and public health experts are essential for addressing the zoonotic transmission cycle, implementing control programs, and educating the public about prevention strategies.

### 4.3. Limitations

While our findings could be considered a substantial contribution to understanding *E. granulossus* in children, several limitations should be noted. (i) Sample size: the relatively small sample size may limit the generalizability of the results. More extensive multicenter studies are needed to validate these findings across diverse pediatric populations. (ii) Lack of long-term follow-up: the absence of long-term follow-up data restricted our ability to assess the recurrence rates and long-term complications. (iii) Potential bias in diagnosis: misdiagnosis is possible due to reliance on imaging and serological tests, especially atypical presentations. (iv) The lack of genetic identification restricts our ability to definitively link contact with intermediate hosts to organ tropism in infected children.

## 5. Conclusions

The present study highlights the complexity of managing *E. granulosus* infections in children. Our findings evidence the importance of a multidisciplinary approach, combining early diagnostic tools, tailored medical therapy, and careful surgical intervention when necessary. An interdisciplinary approach allows for comprehensive patient care while addressing the broader epidemiological and ecological challenges CE poses. For instance, reducing transmission requires the medical treatment of affected individuals and control measures targeting animal reservoirs and definitive hosts. By integrating these perspectives, a multidisciplinary strategy ensures a more holistic and effective response to this disease. Addressing the challenges identified in this research can lessen outcomes and diminish the burden of hydatid disease in pediatric populations. Future directions may explore the pediatric immune response to *E. granulosus* infection in greater detail, developing more sensitive and specific diagnostic markers essential for early detection, especially in asymptomatic cases.

## Figures and Tables

**Figure 1 pathogens-14-00053-f001:**
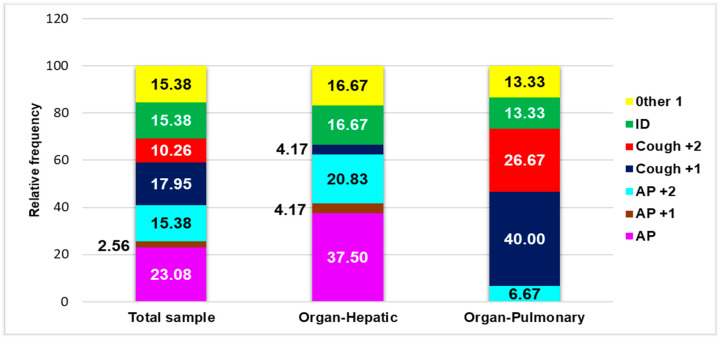
The main symptoms associated with organs involved in CE in pediatric patients. AP = abdominal pain; AP +1 = abdominal pain +1 = abdominal pain with fever; AP +2 = abdominal pain +2 = abdominal pain + fever + headache and abdominal pain + nausea and vomiting; Cough +1 = cough associated with one of the following symptoms: fever, hemoptysis, shortness of breath, or thoracic back pain; Cough +2 = cough + fever and hemoptysis, cough + fever and thoracic pain, cough + shortness of breath and fever; Other 1 = axillary adenopathy, back pain, left thoracic pain, tachycardia, or vomiting; ID = incidental discovery; MD = multiple organ diffusion; C = complications; HD = hospitalization days.

**Figure 2 pathogens-14-00053-f002:**
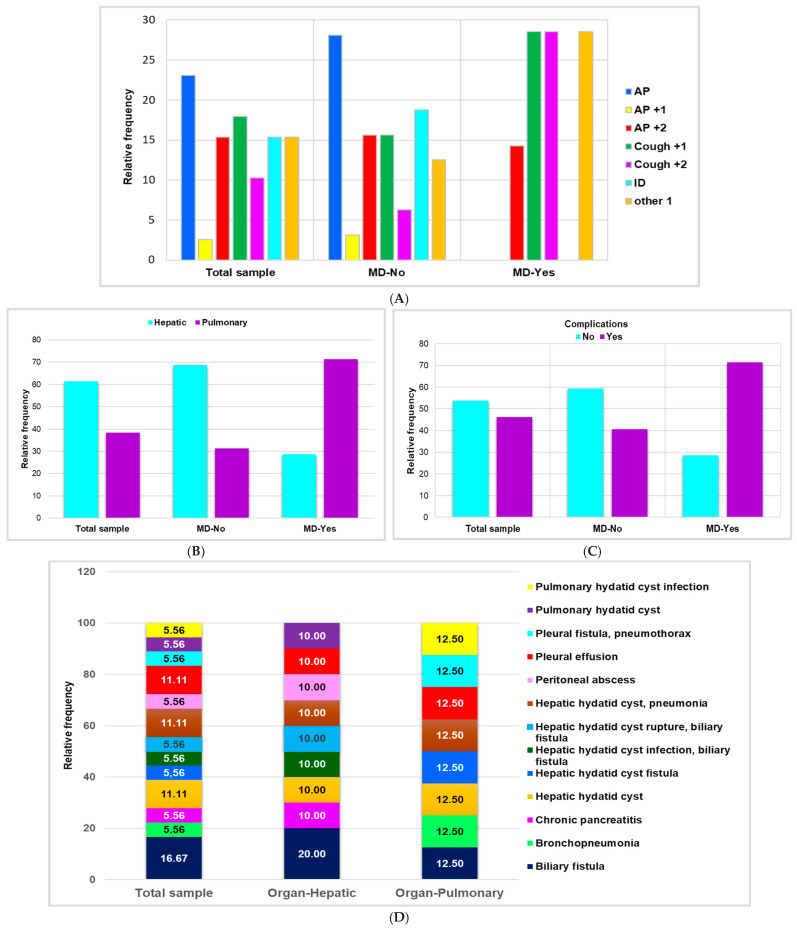
(**A**) The main symptoms associated with MD. (**B**) The correlation between MD and CE localization. (**C**) MD association with complications. (**D**) Complication types. AP = abdominal pain; AP +1 = abdominal pain +1 = abdominal pain with fever; AP +2 = abdominal pain +2 = abdominal pain + fever + headache and abdominal pain + nausea and vomiting; Cough +1 = cough associated with one of the following symptoms: fever, hemoptysis, shortness of breath, or thoracic back pain; Cough +2 = cough + fever and hemoptysis, cough + fever and thoracic pain, cough + shortness of breath and fever; Other 1 = axillary adenopathy, back pain, left thoracic pain, tachycardia, or vomiting; ID = incidental discovery; MD = multiple organ diffusion; C = complications; HD = hospitalization days.

**Figure 3 pathogens-14-00053-f003:**
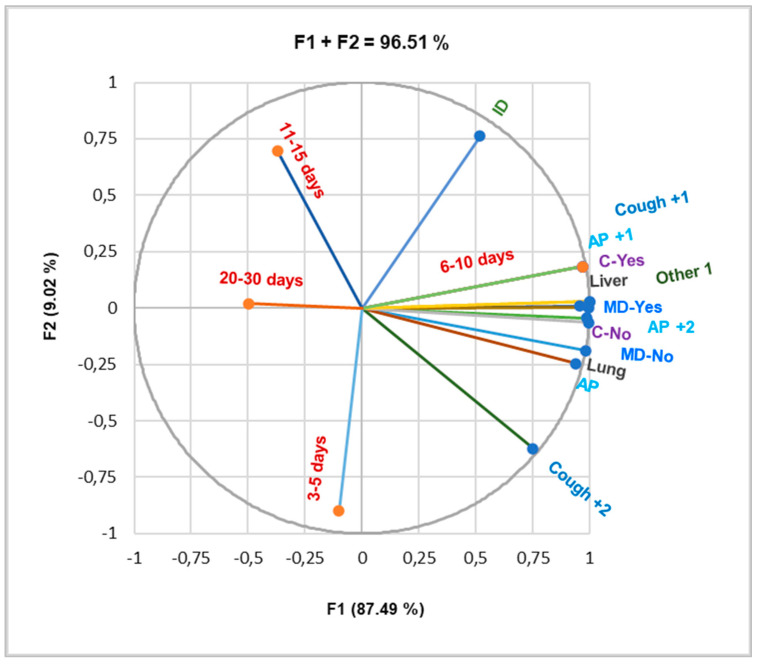
Correlations between the main symptoms at presentation, organ involvement complications, and hospitalization days. AP = abdominal pain; AP +1 = abdominal pain +1 = abdominal pain with fever; AP +2 = abdominal pain +2 = abdominal pain + fever and headache and abdominal pain + nausea and vomiting; Cough +1 = cough associated with one of the following symptoms: fever, hemoptysis, shortness of breath, or thoracic back pain; Cough +2 = cough + fever and hemoptysis, cough + fever and thoracic pain, cough + shortness of breath and fever; Other 1 = axillary adenopathy, back pain, left thoracic pain, tachycardia, or vomiting; ID = incidental discovery; MD = multiple organ diffusion; C = complications.

**Figure 4 pathogens-14-00053-f004:**
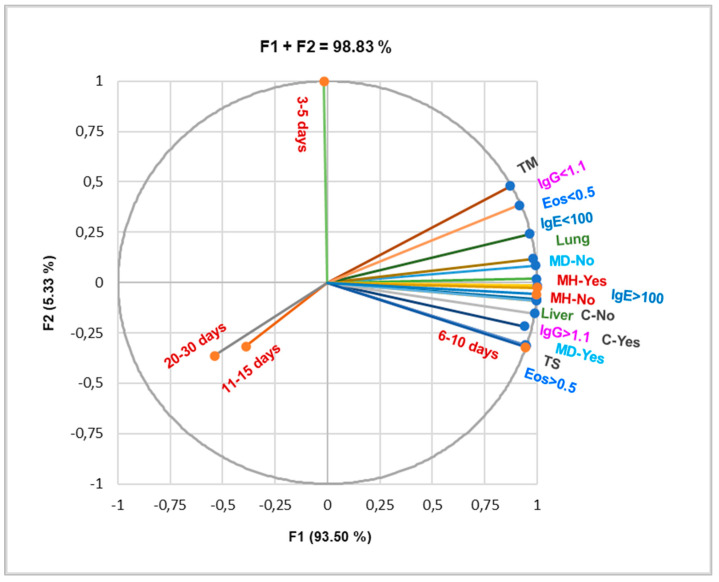
Correlations between organ involvement, laboratory analyses, treatment type, and hospitalization days. MH = multiple hospitalizations; MD = multiple organ diffusion; TM = medication (with albendazole); TS = surgical treatment; Eos = eosinophilia (N/µL); IgE (i.u./mL); IgG (i.u./mL).

**Table 1 pathogens-14-00053-t001:** Sociodemographic data of the pediatric patient group (*n* = 39).

Children 2–15 Years (Average Age = 9.84 Years)		Total		Rural Residence		Urban Residence		*p*-Value
		*n*	%	*n*	%	*n*	%	
Total		39.00	100.00	29.00	74.36	10.00	25.64	<0.05
Sex	Female	15.00	38.46	13.00	44.83	2.00	20.00	<0.05
	Male	24.00	61.54	16.00	55.17	8.00	80.00	
Age	10–15 years	24.00	61.54	17.00	58.62	7.00	70.00	<0.05
	4–9 years	11.00	28.21	9.00	31.03	2.00	20.00	
	≤3 years	4.00	10.26	3.00	10.34	1.00	10.00	
Animal contact	No	11.00	28.21	5.00	17.24	6.00	60.00	<0.05
	Yes	28.00	71.79	24.00	82.76	4.00	40.00	
Animal type	Dog	5.00	17.86	1.00	4.17	4.00	100.00	<0.05
	Goat	9.00	32.14	9.00	37.50	0.00	0.00	
	Pig	5.00	17.86	5.00	20.83	0.00	0.00	
	Sheep	9.00	32.14	9.00	37.50	0.00	0.00	

*n* = number of the pediatric patients (frequency); % = relative frequency (percentage).

**Table 2 pathogens-14-00053-t002:** The main aspects investigated in CE pediatric patients during hospitalization.

Aspect		Total		Days of Hospitalization							
				3–5 Days		6–10 Days		11–15 Days		20–30 Days	
		*n*	%	*n*	%	*n*	%	*n*	%	*n*	%
Total		39	100	9	23.08	22	56.41	5	12.82	3	7.69
Signs	AP	9.00	23.08	3.00	33.33	5.00	22.73	1.00	20.00	0.00	0.00
	AP +1	1.00	2.56	0.00	0.00	1.00	4.55	0.00	0.00	0.00	0.00
	AP +2	6.00	15.38	1.00	11.11	5.00	22.73	0.00	0.00	0.00	0.00
	Cough +1	7.00	17.95	1.00	11.11	4.00	18.18	1.00	20.00	1.00	33.33
	Cough +2	4.00	10.26	2.00	22.22	2.00	9.09	0.00	0.00	0.00	0.00
	ID	6.00	15.38	1.00	11.11	2.00	9.09	2.00	40.00	1.00	33.33
	Other 1	6.00	15.38	1.00	11.11	3.00	13.64	1.00	20.00	1.00	33.33
Organ	Liver	24.00	61.54	5.00	55.56	15.00	68.18	3.00	60.00	1.00	33.33
	Lung	15.00	38.46	4.00	44.44	7.00	31.82	2.00	40.00	2.00	66.67
MD	no	32.00	82.05	8.00	88.89	17.00	77.27	5.00	100.00	2.00	66.67
	yes	7.00	17.95	1.00	11.11	5.00	22.73	0.00	0.00	1.00	33.33
C	no	21.00	53.85	5.00	55.56	13.00	59.09	2.00	40.00	1.00	33.33
	yes	18.00	46.15	4.00	44.44	9.00	40.91	3.00	60.00	2.00	66.67
IgE	IgE < 100	7.00	17.95	2.00	22.22	4.00	18.18	1.00	20.00	0.00	0.00
	IgE > 100	32.00	82.05	7.00	77.78	18.00	81.82	4.00	80.00	3.00	100.00
Eos	Eos < 0.5	26.00	66.67	8.00	88.89	12.00	54.55	3.00	60.00	3.00	100.00
	Eos > 0.5	13.00	33.33	1.00	11.11	10.00	45.45	2.00	40.00	0.00	0.00
IgG	IgG < 1.1	8.00	20.51	3.00	33.33	4.00	18.18	1.00	20.00	0.00	0.00
	IgG > 1.1	31.00	79.49	6.00	66.67	18.00	81.82	4.00	80.00	3.00	100.00
Treatment	TM	11.00	28.21	5.00	55.56	6.00	27.27	0.00	0.00	0.00	0.00
	TS	28.00	71.79	4.00	44.44	16.00	72.73	5.00	100.00	3.00	100.00
MH	no	21.00	53.85	5.00	55.56	11.00	50.00	3.00	60.00	2.00	66.67
	yes	18.00	46.15	4.00	44.44	11.00	50.00	2.00	40.00	1.00	33.33

Signs = main symptoms at presentation in the Emergency Service; AP = abdominal pain; AP +1 = abdominal pain +1 = abdominal pain with fever; AP +2 = abdominal pain +2 = abdominal pain + fever and headache and abdominal pain + nausea and vomiting; Cough +1 = cough associated with one of the following symptoms: fever, hemoptysis, shortness of breath, or thoracic back pain; Cough +2 = cough + fever and hemoptysis, cough + fever and thoracic pain, cough + shortness of breath and fever; Other 1 = axillary adenopathy, back pain, left thoracic pain, tachycardia, or vomiting; ID = incidental discovery; MD = multiple organ diffusion; C = complications; TM = medical treatment (with albendazole); TS = surgical treatment; MH = multiple hospitalizations; Eos = eosinophilia (N/µL); IgE (i.u./mL); IgG (i.u./mL); *n* = number of the pediatric patients (frequency); % = relative frequency (percentage).

**Table 3 pathogens-14-00053-t003:** The main findings in CE pediatric patients: children and adolescents.

Aspect		Age Group					
		10–15 Years		4–9 Years		≤3 Years	
		*n*	%	*n*	%	*n*	%
Sex	F	7.00	29.17	5.00	45.45	3.00	75.00
	M	17.00	70.83	6.00	54.55	1.00	25.00
Residence	Rural	17.00	70.83	9.00	81.82	3.00	75.00
	Urban	7.00	29.17	2.00	18.18	1.00	25.00
Animal contact	Dog	3.00	12.50	2.00	18.18	0.00	0.00
	NA	9.00	37.50	1.00	9.09	1.00	25.00
	Ungulates	12.00	50.00	8.00	72.73	3.00	75.00
Ungulate	Goat	4.00	16.67	3.00	27.27	2.00	50.00
	Pig	2.00	8.33	2.00	18.18	1.00	25.00
	Sheep	6.00	25.00	3.00	27.27	0.00	0.00
Organ	Hepatic	16.00	66.67	7.00	63.64	1.00	25.00
	Pulmonary	8.00	33.33	4.00	36.36	3.00	75.00
MD	MD-No	19.00	79.17	9.00	81.82	4.00	100.00
	MD-Yes	5.00	20.83	2.00	18.18	0.00	0.00
Sign	AP	7.00	29.17	1.00	9.09	1.00	25.00
	AP +1	1.00	4.17	0.00	0.00	0.00	0.00
	AP +2	4.00	16.67	2.00	18.18	0.00	0.00
	Cough +1	2.00	8.33	3.00	27.27	2.00	50.00
	Cough +2	4.00	16.67	0.00	0.00	0.00	0.00
	ID	3.00	12.50	2.00	18.18	1.00	25.00
	Other 1	3.00	12.50	3.00	27.27	0.00	0.00
HD	3–5 days	4.00	16.67	3.00	27.27	2.00	50.00
	6–10 days	16.00	66.67	5.00	45.45	1.00	25.00
	11–15 days	3.00	12.50	2.00	18.18	0.00	0.00
	20–30 days	1.00	4.17	1.00	9.09	1.00	25.00
IgE	IgE < 100	4.00	16.67	2.00	18.18	1.00	25.00
	IgE > 100	20.00	83.33	9.00	81.82	3.00	75.00
Eos	Eos < 0.5	15.00	62.50	8.00	72.73	3.00	75.00
	Eos > 0.5	9.00	37.50	3.00	27.27	1.00	25.00
IgG	IgG < 1.1	5.00	20.83	2.00	18.18	1.00	25.00
	IgG > 1.1	19.00	79.17	9.00	81.82	3.00	75.00
C	C-No	10.00	41.67	8.00	72.73	3.00	75.00
	C-Yes	14.00	58.33	3.00	27.27	1.00	25.00
Treatment	TM	5.00	20.83	5.00	45.45	1.00	25.00
	TS	19.00	79.17	6.00	54.55	3.00	75.00
MH	MH-No	11.00	45.83	9.00	81.82	1.00	25.00
	MH-Yes	13.00	54.17	2.00	18.18	3.00	75.00

F = female; M = male; NA = no contact; Signs = main symptoms at presentation in the Emergency Service; AP = abdominal pain; AP +1 = abdominal pain +1 = abdominal pain with fever; AP +2 = abdominal pain +2 = abdominal pain + fever and headache and abdominal pain + nausea and vomiting; Cough +1 = cough associated with one of the following symptoms: fever, hemoptysis, shortness of breath, or thoracic back pain; Cough +2 = cough + fever and hemoptysis, cough + fever and thoracic pain, cough + shortness of breath and fever; Other 1 = axillary adenopathy, back pain, left thoracic pain, tachycardia, or vomiting; ID = incidental discovery; MD = multiple organ diffusion; C = complications; TM = medication (albendazole); TS = surgical treatment; MH = multiple hospitalizations; Eos = eosinophil level (N/µL); IgE (i.u./mL); IgG (i.u./mL); *n* = number of the pediatric patients (frequency); % = relative frequency (percentage).

## Data Availability

The data presented in this study are available on request from the first author and the first corresponding author due to the ongoing study.

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
