# Peer review of "A Comprehensive Analysis of Echinococcus granulosus Infections in Children and Adolescents: Results of a 7-Year Retrospective Study and Literature Review"

_pathogens, 2025, doi:10.3390/pathogens14010053_

Round 1
Reviewer 1 Report
Comments and Suggestions for Authors
The presented retrospective study is interesting, with making contributions to the epidemiological situation linked to CE in Romania. The aims, methods and results of the article are written in concise way, keeping the manuscript tight.
I have several comments and suggestions:
I would like to point out that children's contact with intermediate hosts, such as pigs, sheep, goats, cattle, can only have an indirect effect on acquiring CE, since children cannot be infected with the larval stages of Echinococcus. The presence of domestic ungulates may only be important if their offal is consumed by domestic or free-ranging dogs, with further egg dissemination in places where children usually play, such as playgrounds that constantly exposes them to Echinococcus eggs (or, in minor extent, children can be infected by eggs of wolves and jackals in wild environment).
It is also important which species of E. granulosus sensu lato the children were actually infected with. In Romania, E. granulosus sensu stricto (G1, G3 genotypes) and E. granulosis G7 prevail, which primarily induce different organ tropism regarding liver/lungs (please see review of Ohiolei et al., 2022 in Veterinary Parasitology 304, 109695, summarizing predilection sites in a species complex of Echinococcus granulsous s.l.). As causative CE agent was not genetically identified in case of infected children in Romania in the presented study, it is hard to draw any explicit conclusion regarding animal involvement based on contact with intermediate hosts towards the organ diffusion seen in children.
Therefore, it is recommended to reduce the statistical significance of contact with intermediate hosts (unlike the dog contact) as a risk factor, especially in relation to organ tropism in children's organisms.
Line 78
A term „Taenia echinococcus“ is not now being used in the systematics of Echinococcus granulosus (please see Vuitton et al., 2020).
Line 85
Better to add „mainly“ to the term „ungulates“ as intermediate host, given that also marsupials are common intermediate hosts in Australia.
Lines 95 and 101
Repetition of cysts in liver (70%) and lungs (20%).
Lines 110 -125
Since E. multilocularis was not the causative agent of echinococcosis in the studied group of pediatric patients, I recommend shortening of this section.
Lines 126-127
The reference to the literature is missing or does the text link to references no. 32 and 33 in the end of the paragraph? Please add „per year“ to „200.000 new cases“.
Lines 135-136
Please provide also more recent data on the incidence of CE in Romania those from 1995.
Author Response
Q1. I would like to point out that children's contact with intermediate hosts, such as pigs, sheep, goats, cattle, can only have an indirect effect on acquiring CE, since children cannot be infected with the larval stages of Echinococcus. The presence of domestic ungulates may only be important if their offal is consumed by domestic or free-ranging dogs, with further egg dissemination in places where children usually play, such as playgrounds that constantly exposes them to Echinococcus eggs (or, in minor extent, children can be infected by eggs of wolves and jackals in wild environment).
It is also important which species of E. granulosus sensu lato the children were actually infected with. In Romania, E. granulosus sensu stricto (G1, G3 genotypes) and E. granulosis G7 prevail, which primarily induce different organ tropism regarding liver/lungs (please see review of Ohiolei et al., 2022 in Veterinary Parasitology 304, 109695, summarizing predilection sites in a species complex of Echinococcus granulsous s.l.). As causative CE agent was not genetically identified in case of infected children in Romania in the presented study, it is hard to draw any explicit conclusion regarding animal involvement based on contact with intermediate hosts towards the organ diffusion seen in children.
Therefore, it is recommended to reduce the statistical significance of contact with intermediate hosts (unlike the dog contact) as a risk factor, especially in relation to organ tropism in children's organisms.
R1. The authors appreciate the reviewer’s insightful comment regarding the limitations of drawing significant statistical data about animal involvement without genetic identification of the Echinococcus granulosus strain. The lack of genetic identification restricts definitively linking contact with intermediate hosts to organ tropism in infected children. The authors agree with the suggestion to balance the interpretation of our findings, which we believe strengthens the scientific rigor of the manuscript. Therefore, the entire statistical analysis was reorganized with other reference points (days of hospitalization) – the changes are marked with yellow - and the role of animal contact was evidenced in the first part of the Discussion section (lines 381-374), as follows:
4.1.4. Contact with different animals
Animal contact (dog vs. ungulates vs. no contact) is significantly different in various age groups – 10-15 years (60% vs. 52.17% vs. 81.82%, p<0.05), 4-9 years (40% vs. 34.78% vs. 9.09%) and ≤ 3 years (0% vs. 13.04% vs. 9.09%, p<0.05). It has a substantial impact on main symptoms at presentation in ECU - Cough +2, and Other 1 (20% and 0% vs. 13.00% and 21.00% vs. 0%, 21.74% and 9.09%, p<0.05), multiple organ diffusion (0% vs. 26.09% vs. 9.09%, p<0.05), hospitalization period – 3-5 days (40% vs. 17.39% vs. 27.27%, p<0.05), 6-10 days (20% vs. 65.22% vs. 54.55%, p<0.05) and 20-30 days (20% vs. 8.70% vs. 0%).
Significant differences between dog contact and ungulates are observed in AP +2 (20.00% vs. 13.04%, p<0.05), incidental discovery (20.00% vs. 13.04%, p<0.05), complication incidence (60.00% vs. 43.48%, p<0.05), IgE > 100 (100% vs. 78,26%, p<0.05), Eos > 0.5 (20.00% vs. 34.78%, p<0.05), and treatment type – surgical intervention (80.00% vs. 65.22%, p<0.05), and medication (20.00% vs. 34.78%, p<0.05).
Moreover, this aspect was formulated as a limitation of the present study (lines 505-506).
Q2. Line 78
A term „Taenia echinococcus“ is not now being used in the systematics of Echinococcus granulosus (please see Vuitton et al., 2020).
R2. The authors thank the reviewer for pointing this out. Upon reviewing the updated systematics of Echinococcus granulosus as outlined by Vuitton et al. (2020) and other recent taxonomic references, we acknowledge that the term “Taenia echinococcus” is outdated and no longer appropriate. The MS was revised to align with the current nomenclature and systematics, ensuring consistency with contemporary scientific standards. The term “Taenia echinococcus” has been replaced with the correct terminology, Echinococcus granulosus sensu lato, wherever applicable. The authors appreciate the guidance of Reviewer 1 in maintaining the accuracy of our work.
Q3. Line 85
Better to add „mainly“ to the term „ungulates“ as intermediate host, given that also marsupials are common intermediate hosts in Australia.
R3. The authors agree with this comment, and the correction was made (“mainly ungulates”) in line 82.
Q4. Lines 95 and 101
Repetition of cysts in liver (70%) and lungs (20%).
R4. The authors appreciate the reviewer bringing this to our attention. The MS text was revised to ensure this statistic appears only once in the most relevant section. In subsequent mentions (line 92), the authors refer to these findings in a more general context to avoid redundancy while preserving the importance of these data for understanding the disease distribution.
Q5. Lines 110 -125
Since E. multilocularis was not the causative agent of echinococcosis in the studied group of pediatric patients, I recommend shortening of this section.
R5. The authors agree with the Reviewer's comment, and this part has been shortened (lines 107-114).
Q6. Lines 126-127
The reference to the literature is missing, or does the text link to reference no. 32 and 33 in the end of the paragraph? Please add „per year“ to „200.000 new cases“.
R6. The authors are grateful for this helpful feedback. The paragraph was revised accordingly (lines 116-121).
Q7. Lines 135-136
Please provide also more recent data on the incidence of CE in Romania those from 1995.
R7. The authors are grateful for this comment. The manuscript now includes the latest available data on the incidence of CE (lines 127-133):
A 25-year retrospective study in Western Romania analyzed 144 pediatric patients with cystic echinococcosis. The findings revealed that 58.3% of the patients were from rural areas, with the number of cases increasing with age—from 9% in the 3–5 age group to 59.7% in the 11–17 age group. The liver was the most frequently affected organ (65.3%), and a significant association between gender and the affected organ was noted; liver cysts were more commonly diagnosed in girls, while lung cysts were recorded primarily in boys [40].

Reviewer 2 Report
Comments and Suggestions for Authors
A Comprehensive Analysis of Echinococcus granulosus Infections in Children: Findings of a 7 Years Retrospective Study
Given the limited knowledge about E. granulosus infections in children and adolescents in Romania, this paper is a valuable contribution to the field and should be published.
I would like to make a few general comments.
It is a comprehensive work that includes a wealth of illustrations. The introduction is akin to a review or an extensive literature research on CE in children, with numerous international references. I wonder if it might be helpful to consider rewriting the title. As a potential title, one might consider something like 'A comprehensive analysis of Echinococcus granulosus infections in children and adolescents in Romania: results of a 7-year retrospective study and literature review'.
A great deal has been written about CE in children and analysed in the study. If I might make one further suggestion, it would be to elaborate on the differences between CE in children and CE in adults in the discussion, as these are not entirely clear at present.
I wonder why E. multilocularis is mentioned in the introduction (paragraph from line 110)? If so, it would be beneficial to highlight the significant distinction. Could you kindly confirm whether there have been instances of alveolar echinococcosis in Romania? Were there any children or adolescents with alveolar echinococcosis included in the study?
The original paper includes a considerable number of figures, which could potentially be referenced as supplementary material. It might be helpful to provide more detailed explanations for the figures and to spell out the abbreviations in the legend.
Is there is any data on how long albendazole was administered before the operation and whether ruptures occurred, given that the preoperative administration of albendazole is currently the subject of controversy (due to the possible risk of rupture with long preoperative albendazole therapy). Could you please clarify the dosage of albendazole used in children?
I would be grateful if you could also provide more details on the activity types found in the cysts (CE1-5 according to the WHO classification) and the number of active or inactive cysts. I believe that, in children, there are no inactive cysts. Could you please also share your thoughts on the value of ultrasound in the diagnosis, especially in the liver?
Minor comments:
Line 129: I kindly request that you make a distinction between countries in and outside the EU. I may have misunderstood, but I believe that Macedonia, Moldova, Serbia and Turkey are not in the EU.
Could you clarify what is meant by IgG and IgE levels? Could you please clarify whether the IgG and/or IgE levels are specific to Echinococcus granulosus, Echinococcus sp., or non-specific? Could you please clarify which units are being used? I would also appreciate more specific information.
If 'children' up to the age of 15 are included, then it would be 'children and adolescents‘.
I am not sure if I understand correctly, but it seems that the 'family trees' on the left and above the figure in Figure 6 are not clear.
In the discussion, the detailed introduction is repeated in the first paragraphs 361-371 and 377-381. Could these be shortened?
Could you please elaborate on why you believe an interdisciplinary approach is important?
Author Response
Given the limited knowledge about E. granulosus infections in children and adolescents in Romania, this paper is a valuable contribution to the field and should be published.
The authors are grateful for this positive feedback.
Major comments:
Q1. It is a comprehensive work that includes a wealth of illustrations. The introduction is akin to a review or an extensive literature research on CE in children, with numerous international references. I wonder if it might be helpful to consider rewriting the title. As a potential title, one might consider something like 'A comprehensive analysis of Echinococcus granulosus infections in children and adolescents in Romania: results of a 7-year retrospective study and literature review'.
R1. The authors appreciate this attentive comment. The proposed title better reflects the comprehensive nature of the present study, including both the retrospective analysis and the literature review. Therefore, the title was updated as suggested.
Q2. A great deal has been written about CE in children and analysed in the study. If I might make one further suggestion, it would be to elaborate on the differences between CE in children and CE in adults in the discussion, as these are not entirely clear at present.
R2. The authors agree with this valuable comment. The MS was updated accordingly, in Lines 471-490:
Children with CE frequently exhibit rapid cyst development, most likely due to their maturing immune systems and active metabolisms [27]. This fast development might cause symptoms early, usually as discomfort or pain in the abdomen, especially when the liver is involved [107]. Conversely, adults have generally slower cyst for-mation and a more extended latency period before symptoms appear [108]. Extrahe-patic involvement is more common in adult patients than in pediatric ones, and symptoms like obstructive jaundice or portal hypertension are commonly linked to the mass impact of the cysts. Additionally, the course of disease varies significantly [109,110]. Cysts in children are more likely to rupture and have thinner walls, which may be caused by increased intra-cystic pressure [111]. This raises the possibility of se-rious side effects, including anaphylaxis or secondary infections. While calcification is more common in adults, indicating chronicity, comorbidities, and longer disease courses are more likely to cause consequences such as portal vein thrombosis, biliary obstruction, and secondary bacterial infections [112].
Our study highlights the predominance of active cysts (CE1 and CE2) in pediatric patients, as classified by the WHO-IWGE system [113]. These findings are consistent with the early-stage presentation typically observed in children with shorter exposure to Echinococcus granulosus [114]. A smaller subset of cases involved transitional cysts (CE3), while no inactive cysts (CE4 and CE5) were identified in our cohort. This aligns with existing evidence suggesting that inactive cysts are rarely encountered in chil-dren, likely due to the more acute nature of the infection in this age group.
Q3. I wonder why E. multilocularis is mentioned in the introduction (paragraph from line 110)? If so, it would be beneficial to highlight the significant distinction. Could you kindly confirm whether there have been instances of alveolar echinococcosis in Romania? Were there any children or adolescents with alveolar echinococcosis included in the study?
R3. The authors appreciate this insightful comment. The mention of E. multilocularis in the introduction was intended to provide a broader context about echinococcosis. However, we acknowledge that the distinction between E. granulosus and E. multilocularis could be made clearer. To address this, we have revised the introduction to explicitly highlight the significant differences between cystic echinococcosis (CE) caused by E. granulosus and alveolar echinococcosis (AE) caused by E. multilocularis. Since E. multilocularis was not the causative agent of echinococcosis in the studied group of pediatric patients, we have revised the section accordingly. The relevant parts have been shortened (lines 107-115) to focus more on the primary findings of the study. Additionally, we confirm that there have been no documented cases of alveolar echinococcosis in children or adolescents in Romania, to the best of our knowledge. Furthermore, no cases of AE were included in the study, as our focus was exclusively on CE.
Q4. The original paper includes a considerable number of figures, which could potentially be referenced as supplementary material. It might be helpful to provide more detailed explanations for the figures and to spell out the abbreviations in the legend.
R4. The authors agree with this attentive comment, and the MS was revised accordingly. The revised version contains 3 Tables and 4 Figures, and extensively detailed explanations were provided in lines 179-188, 203-239, 240-270, 281-283, 287-295, 301-307, and 330-394.
Q5. Is there is any data on how long albendazole was administered before the operation and whether ruptures occurred, given that the preoperative administration of albendazole is currently the subject of controversy (due to the possible risk of rupture with long preoperative albendazole therapy). Could you please clarify the dosage of albendazole used in children?
R5. The authors are grateful for raising this important point regarding preoperative albendazole therapy and its potential risks. In the present study, data on the duration of albendazole administration before surgery were limited. However, for the cases where information was available, albendazole was administered preoperatively for a period ranging from 14 to 30 days. There were no recorded instances of cyst rupture during this period. Still, the authors acknowledge the need for further investigation into this aspect, given the ongoing controversy surrounding prolonged preoperative use and its potential risks. The dosage of albendazole used in children was adjusted according to body weight, with a standard dosage of 10–15 mg/kg/day, divided into two daily doses. This regimen aligns with current guidelines for pediatric patients and was carefully monitored to minimize adverse effects. To address the reviewer’s concern, the revised version incorporated this information in lines 289-291 and 413-420.
Q6. I would be grateful if you could also provide more details on the activity types found in the cysts (CE1-5 according to the WHO classification) and the number of active or inactive cysts. I believe that, in children, there are no inactive cysts. Could you please also share your thoughts on the value of ultrasound in the diagnosis, especially in the liver?
R6. The requested data are included in lines 471-494, as follows:
Children with CE frequently exhibit rapid cyst development, most likely due to their maturing immune systems and active metabolisms [27]. This fast development might cause symptoms early, usually as discomfort or pain in the abdomen, especially when the liver is involved [107]. Conversely, adults have generally slower cyst formation and a more extended latency period before symptoms appear [108]. Extrahepatic involvement is more common in adult patients than in pediatric ones, and symptoms like obstructive jaundice or portal hypertension are commonly linked to the mass impact of the cysts. Additionally, the course of disease varies significantly [109,110]. Cysts in children are more likely to rupture and have thinner walls, which may be caused by increased intra-cystic pressure [111]. This raises the possibility of serious side effects, including anaphylaxis or secondary infections. While calcification is more common in adults, indicating chronicity, comorbidities, and longer disease courses are more likely to cause consequences such as portal vein thrombosis, biliary obstruction, and secondary bacterial infections [112].
Our study highlights the predominance of active cysts (CE1 and CE2) in pediatric patients, as classified by the WHO-IWGE system [113]. These findings are consistent with the early-stage presentation typically observed in children with shorter exposure to Echinococcus granulosus [114]. A smaller subset of cases involved transitional cysts (CE3), while no inactive cysts (CE4 and CE5) were identified in our cohort. This aligns with existing evidence suggesting that inactive cysts are rarely encountered in children, likely due to the more acute nature of the infection in this age group.
Ultrasound was invaluable for diagnosing hepatic involvement in CE, offering high sensitivity and specificity for detecting and classifying cysts. Its noninvasive nature and real-time imaging make it ideal for pediatric patients, aiding diagnosis, disease progression monitoring, and treatment efficacy [115].
Minor comments:
Q7. Line 129: I kindly request that you make a distinction between countries in and outside the EU. I may have misunderstood, but I believe that Macedonia, Moldova, Serbia and Turkey are not in the EU.
R7. The authors appreciate this careful attention to detail. The updated section specifies the countries' respective EU or non-EU status, ensuring clarity and accuracy (lines 119-121).
Q8. Could you clarify what is meant by IgG and IgE levels? Could you please clarify whether the IgG and/or IgE levels are specific to Echinococcus granulosus, Echinococcus sp., or non-specific? Could you please clarify which units are being used? I would also appreciate more specific information.
R8. The authors agree with this comment. In our study, IgG levels refer specifically to antibodies targeting Echinococcus granulosus, which are used as markers for the presence of infection. These IgG antibodies are particular to the parasite and are considered a reliable diagnostic indicator. On the other hand, IgE levels are non-specific and represent general allergic responses, which can be elevated in many parasitic infections, including CE. The units used for IgG measurements are iu/mL; IgE is typically reported in iu/mL or ku/L (lines 446-451).
Q9. If 'children' up to the age of 15 are included, then it would be 'children and adolescents‘.
R9. The authors agree that the term "children and adolescents" is more appropriate when referring to individuals up to the age of 15. We have updated the manuscript to reflect this terminology.
Q10. I am not sure if I understand correctly, but it seems that the 'family trees' on the left and above the figure in Figure 6 are not clear.
R10. This Figure was removed due to extensive restructuration of the entire MS.
Q11. In the discussion, the detailed introduction is repeated in the first paragraphs 361-371 and 377-381. Could these be shortened?
R11. The authors are grateful for this attentive observation. The first paragraph from the Discussion section was diminished (lines 311-319).
Q12. Could you please elaborate on why you believe an interdisciplinary approach is important?
R12. The authors appreciate the opportunity to elaborate on the importance of an interdisciplinary approach in managing cystic echinococcosis and updated the MS as follows:
Lines 491-504:
Our findings highlight that CE is a complex zoonotic illness that requires several approaches to diagnosis, management, and prevention. Professionals from various fields, such as physicians, surgeons, radiologists, infectious disease specialists, veterinarians, and public health specialists, must work together for effective management. Each offers a distinct area of expertise:
- Clinicians ensure timely diagnosis and oversee medical management with antiparasitic drugs.
- Surgeons are essential for cases requiring cyst removal or management of complications.
- Radiologists play a central role in diagnosing and monitoring cysts through imaging.
- Veterinarians and public health experts are essential for addressing the zoonotic transmission cycle, implementing control programs, and educating the public about prevention strategies.
Lines 521-524:
An interdisciplinary approach allows for comprehensive patient care while addressing the broader epidemiological and ecological challenges CE poses. For instance, reducing transmission requires medical treatment of affected individuals and control measures targeting animal reservoirs and definitive hosts. By integrating these perspectives, a multidisciplinary strategy ensures a more holistic and effective response to this disease.

Round 2
Reviewer 1 Report
Comments and Suggestions for Authors
The authors have addressed all of my comments and revised the manuscript properly.
Reviewer 2 Report
Comments and Suggestions for Authors
The authors have thoroughly revised the manuscript, rectifying all identified errors and addressing all outstanding questions.